# Classification and Expression Profile of the U-Box E3 Ubiquitin Ligase Enzyme Gene Family in Maize (*Zea mays* L.)

**DOI:** 10.3390/plants11192459

**Published:** 2022-09-21

**Authors:** Xiangnan Li, Longming Zhu, Zhenxing Wu, Jianjian Chen, Tingzhen Wang, Xiaoli Zhang, Gaofu Mei, Jian Wang, Guihua Lv

**Affiliations:** 1Institute of Maize and Featured Dryland Crops, Zhejiang Academy of Agricultural Sciences, Dongyang 322100, China; 2Institute of Crops and Nuclear Technology Utilization, Zhejiang Academy of Agricultural Sciences, Hangzhou 310021, China; 3Central Laboratory of Zhejiang, Academy of Agricultural Sciences, Hangzhou 310021, China

**Keywords:** maize (*Zea mays* L.), U-box E3 (*PUB*) gene family, classification, expression profile

## Abstract

The U-box E3 (*PUB*) family genes encode the E3 ubiquitin ligase enzyme, which determines substrate specific recognition during protein ubiquitination. They are widespread in plants and are critical for plant growth, development, and response to external stresses. However, there are few studies on the functional characteristic of *PUB* gene family in the important staple crop, maize (*Zea mays* L.). In this study, the *PUB* gene in maize was aimed to identify and classify through whole-genome screening. Phylogenetic tree, gene structure, conserved motif, chromosome location, gene duplication (GD), synteny, and cis-acting regulatory element of *PUB* member were analyzed. The expression profiles of *ZmPUB* gene family in maize during development and under abiotic stress and hormones treatment were analyzed by the RNA-seq data. A total of 79 *PUB* genes were identified in maize genome, and they were stratified into seven categories. There were 25 pairs of segmental duplications (SD) and 1 pair of tandem duplication (TD) identified in the maize PUB gene family. A close relationship was observed between the monocot plant maize and rice in *PUB* gene family. There were 94 kinds of cis-acting elements identified in the maize *PUB* gene family, which included 46 biotic- and abiotic-responsive elements, 19 hormone-responsive elements, 13 metabolic and growth-related elements. The expression profiles of maize *PUB* gene family showed characteristics of tissue specificity and response to abiotic stress and hormones treatment. These results provided an extensive overview of the maize *PUB* gene family.

## 1. Introduction

The ubiquitin proteasome system (UPS) is an energy dependent system that modulates protein activity and degeneration. It is involved in cellular growth, differentiation and apoptosis, secretion and endocytosis, gene transcription and expression, as well as signal transduction and immune response [1,2]. UPS is composed of the ubiquitin-activating (E1), ubiquitin-conjugating (E2), and ubiquitin ligase (E3) enzymes. Ubiquitination is a common post-translational modification involving proteins. The process is as follows: E1 binds ubiquitin using a high-energy, thiol-ester bond in an ATP-based system. The E1-ubiquitin complex then interacts with E2 and transfers the active ubiquitin to a cysteine residue in E2. E3 next recognizes the substrate proteins and catalyzes the formation of an isopeptide bond between the C-terminal carboxyl group of ubiquitin and free lysl-amino groups within the protein substrate [3,4,5,6,7]. Finally, the ubiquitinated protein is degraded by 26S protease [1].

The ubiquitin ligase enzymes (E3) determine ubiquitin-specificity via recognition of substrate proteins, and they constitute the largest family among the three enzymes. According to the functional mechanism and subunit composition, E3 can be classified into four distinct categories: homologous to E6-AP COOH-terminus (HECT), U-box, really interesting new gene (RING), and cullin-RING ligase (CRLs) [8]. The U-box E3 (PUB) ligase enzymes contain the U-box domain, which are made up of approximately 70 amino acids. Moreover, they are highly conserved in eukaryotes, such as plants, animals, and yeast [9]. The prototype U-box protein was initially reported in yeast [10]. Subsequently, the U-box protein was also recognized in mammals, wherein it influences ubiquitination in combination with E1 and E2 and in the absence of additional E3 components [11]. To date, the *PUB* gene family was detected in numerous species, namely Arabidopsis (*Arabidopsis thaliana*, 64), rice (*Oryza sativa*, 77), Chinese cabbage (*Brassica rapa* ssp. *Pekinesis,* 101), soybean (*Glycine max*, 125), barley (*Hordeum vulgare*, 67), tomato (*Lycopersicon esculentum*, 62), citrus (*Citrus clementina*, 56), grape (*Vitis vinifera*, 56), cotton (*Gossypium raimondii*, 93), and Medicago (*Medicago*
*truncatula*, 41) [12,13,14,15,16,17,18,19,20,21].

The *PUB* gene family modulates protein ubiquitination and degradation, and it plays essential roles in growth, development, reproduction, biotic and abiotic stress, as well as hormones [1,11,22,23]. In rice, *OsPUB75* encodes an active PUB ligase enzyme. A mutation in this gene decreases cellular proliferation, disorganizes cellular files in aerial organs, and eventually causes the dwarf phenotype [24]. In barley, the *brh2* and *ari-l* mutant exhibits PUB ligase enzyme deficiency and produces a relatively strong semi-dwarf phenotype [25]. In Arabidopsis, *AtPUB4* serves as a global modulator of cellular proliferation and division, which are critical elements of the root architecture [26]. In Brassica, *ARC1* promotes the ubiquitination and proteasomal destruction of compatibility factors within the pistil, thereby resulting in pollen rejection [27]. The PUB gene family is also known to modulate abiotic and biotic stress responses as well. The PUB ligase enzyme CMPG1 is critical for plant defense and disease resistance in tobacco (*Nicotiana tabacum*) and tomato, and it is homologous to the encoding proteins of *AtPUB20* and *AtPUB21* in Arabidopsis [28]. *GmPUB1* is up-regulated after *Phytophthora sojae* infection, and *GmPUB1* gene silencing in soybean produces a loss of race-specific *Phytophthora* resistance [29]. Arabidopsis *AtPUB30* participates in salt-stress tolerance as a negative modulator of the germination stage in root tissues [30]. *OsPUB67* strongly regulates drought tolerance in rice, and over-expression of *OsPUB67* enhances drought-stress tolerance by augmenting reactive oxygen scavenging capability and closing the stomata [31]. *VaPUB* encodes a novel PUB ligase enzyme, which is upregulated by cold stress. Overexpression of *VaPUB* augments cold- and salt-stress tolerance [19]. Arabidopsis *AtPUB19* is highly up-regulated by salt, drought, cold, and heat stress. Knock out of *AtPUBI9* substantially decreased the resistance to high temperature but enhanced the resistance to drought [32]. In addition, there are multiple *PUB* genes that play distinct roles in hormonal response. *AtPUB10* negatively regulates the abscisic acid (ABA) in Arabidopsis [33]. In rice, a *DSG1*-encoded PUB ligase enzyme is negatively modulated by brassinosteroid (BR), ethylene (ETH), auxin (AUX), and salicylic acid (SA), and its mutant is less responsive to brassinosteroid compared to the wild-type rice [34]. The PUB ligase enzyme PHOR1 serves an essential function in the gibberellin (GA) response, and its function is conserved in potato (*Solanum tuberosum*) and Arabidopsis [35,36].

The *PUB* gene family serves essential functions in regulating plant development as well as response to biotic, abiotic, and hormonal stressors. However, the current published studies were primarily focused on model plants such as Arabidopsis, cotton, tomato, rice, and Medicago. Hence, the examination of the *PUB* gene family in maize is rather insufficient, particularly the assessment of its gene expression profile. Herein, we aimed to screen and classify the *PUB* gene family in maize and performed systematic and comprehensive analyses. This study would provide a basis for the in-depth study of the physiological activities associated with these genes.

## 2. Results

### 2.1. Identification and Phylogenetic Analysis of PUB Genes in Maize

Overall, 79 putative *PUB* genes were screened with HMMER, using default parameters and significant e*^−^*^3^ value against the maize genome. We named these genes as *ZmPUB1* to *ZmPUB79*. According to Pfam and SMART, all 79 *ZmPUBs* contained the U-box domain (PF04564). We further analyzed the gene locus, chromosomal location, sequence length, exon amount, molecular weight (MW), isoelectric point (PI), and subcellular location of all *ZmPUB* genes (Appendix A).

The protein length of *ZmPUBs* was between 94–1353 amino acids, and the MW was between 10.75–144.72 kDa. The PI range was 4.57 to 9.80. The subcellular localization was estimated using the WoLF PSORT website. Among the 79 ZmPUB proteins, 19 were predicted to be cytoplasmic proteins; 14 were located in the nucleus; 10 were in the plasma membrane; and the rest were localized in various organelles, such as the chloroplast (29), endoplasmic reticulum (3), golgi apparatus (2), mitochondrion (1), and peroxisome (1).

To explore the relationship among different *PUB* genes, we generated a neighbor-joining phylogenetic tree with the U-box domain sequences of ZmPUBs, AtPUBs, and OsPUBs. Based on our analysis, the *79* ZmPUB proteins were stratified into seven categories, namely group I to group VII (Figure 1). Group IV consisted of the most members, namely 20 ZmPUB proteins, followed by group VII (19), and group II (17). Groups I and V possessed the least amounts of members, including only four and five ZmPUB proteins, respectively. The protein structure domain analysis was carried out using SMART and Pfam databases (Appendix A). All ZmPUB proteins harbored the U-box domain. From group I to group VII, the proportion of ZmPUB proteins containing only the U-box domain was 50% (2/4), 11.7% (2/17), 42.8% (3/7), 35% (7/20), 100% (5/5), 0 (0/7), and 89.4% (17/19). We also identified an additional 10 structural domains in the ZmPUB proteins. In group I, the RS4NT or TPR domain was detected in two ZmPUB proteins. In groups II and IV, a total of eight domain types were observed, including the ARM, pkinase-tyr, coil coil region, UFD2P_CORE, USP, TPR, WD40, and KAP domains. In groups V, VI, and VII, there were three major domain types, which were all in the coil coil region, as well as the ZnF_TTF and ARM domains, and they were located in 19 ZmPUB proteins. In maize, most ZmPUB proteins carried only the U-box domain, accounting for 45.6% (36/79) of total proteins. There were similar proportions surveyed in Arabidopsis, rice, barley and citrus, and they were 25.0% (16/64), 25.9% (20/77), 31.3% (21/67), and 30.3% (17/56), respectively [12,13,18,20]. A previous study revealed that the ARM domain was related to substrate recognition during the ubiquitination process, and it is the second populated domain in rice and Arabidopsis [21]. In maize, 25 ZmPUB proteins harbored the ARM domain, and it was similar to rice and Arabidopsis. Additionally, approximately 70–74% of ZmPUB proteins were clustered into three groups (groups II, IV, and VII) in rice, Arabidopsis, and maize (Figure 1). This suggested that the maize *PUB* gene family was evolutionarily conserved.

### 2.2. Genetic Structures and Conserved Motifs Analyses of the PUB Gene Family

To better elucidate the association between gene function and evolution, we explored the structural organization and conserved motifs of *PUB* genes (Figure 2, Appendix A). We observed between 1 to 16 exons, with a mean of 3.61 exons per gene. There were about 37.9% *ZmPUB* genes with one exon but no intron. The proportions of *ZmPUB* genes containing two, three, and four exons were 9.8%, 8.8%, and 13.5%, respectively. The largest number of exons was 16, and they were detected in *ZmPUB4* and *ZmPUB69*. In group I, the exon number in *ZmPUB* genes was between 5–8, with a mean of 6.75. In groups II and IV, we detected the maximum variation of exons, which was 1–16 and 2–9, respectively. In group III, there were 2–4 exons per gene. Around 84–100% of *ZmPUB* genes from groups V, VI, and VII harbored only one exon. The *ZmPUB* genes from the same group shared comparable gene structure, thereby suggesting functional conservation among the maize *PUB* gene family. Additionally, the exon variations in *ZmPUB* genes may indicate the function diversity of the maize *PUB* gene family.

We screened ten conserved motifs among the 79 *ZmPUB* genes (Figure 2, Appendix A). About 83% of *ZmPUB* genes contained motifs 1, 3 and 4 simultaneously. Motifs 6, 7, and 9 mainly existed in the *ZmPUB* genes of group II. The tandem repeats of motifs 2 and 5 were characteristic of *ZmPUB* genes in group IV, and it likely served a distinct biological function. Motif 8 was prevalent exclusively among groups IV and VI. Motif 10 existed in all groups except for groups I and II. The conserved motif analysis suggested that the *PUB* genes reserved the U-box domain and accumulated additional motifs over the course of evolution. Lastly, similar motifs among *ZmPUB* genes might suggest the conserved evolutionary relationship and similar biological function.

### 2.3. Chromosomal Locations, Duplications, and Synteny Analysis of the Maize PUB Gene Family

To examine the chromosomal distribution of *ZmPUB* genes, we screened the genomic database, based on the DNA sequence of individual *ZmPUB* genes, and drew the chromosomal location map via the MapChart software (Figure 3). In total, 77 *ZmPUB* genes (97.4%) were mapped to 10 chromosomes, and they were unevenly distributed within each chromosome. We identified 10, 12, and 13 *ZmPUB* genes at chromosomes 1, 4, and 5, respectively. On chromosomes 2, 3, 9, and 10 were identified 9, 8, 8, and 7 *ZmPUB* genes, respectively. Lastly, only two *ZmPUB* genes were mapped to the chromosome 8.

Gene duplication (GD) involves segmental duplication (SD) and tandem duplication (TD), and it promotes the expansion of the gene family. We conducted GD analysis to reveal the expansion process of the maize *PUB* genes (Figure 4). We identified 25 pairs (43 *ZmPUB* genes) of SD and one pair of TD (*ZmPUB43*/*ZmPUB44*) (Figure 3 and Figure 4). GD occurred on one or two loci. Using synteny analysis, we identified eight *ZmPUB* genes (*ZmPUB27, 32, 43, 45, 48, 55, 58* and *62*) with duplications on two loci, whereas the rest of the genes exhibited duplication on one locus (Figure 4). To elucidate the evolutionary pressures acting on *ZmPUB* genes, we computed the Ka and Ks of the duplicated gene pair, which represented nonsynonymous and synonymous substitution rates, respectively (Table 1). The Ka values were between 0.02–1.00. The Ks values were between 0.17–2.81. The Ka/Ks values of 26 gene pairs were between 0.05 and 0.99, which suggested that these *ZmPUB* genes evolved under strong purifying selection (Ka/Ks < 1). In addition, we computed the probable dates of the duplication events, and they occurred between 13.06 Mya (Ks = 0.17, Million years ago) and 215.77 Mya (Ks = 2.81, Million years ago), with an average of 93.55 Mya (Ks = 1.21, Million years ago). To further explore the evolutionary associations of the *PUB* genes in Arabidopsis, maize, and rice, we assessed syntenic interactions among the three species using the software MCScanX (Figure 5). A total of 64, 79, and 77 *PUB* genes were used from Arabidopsis, maize, and rice, respectively [12,13]. There were 53 syntenic *PUB* gene pairs screened between maize and rice, while only 5 syntenic *PUB* gene pairs were found in Arabidopsis and maize.

### 2.4. Cis-Acting Regulatory Element Analysis

To better elucidate the transcriptional modulatory pathways associated with *ZmPUB* genes, we performed cis-acting regulatory element analysis with the 1500 bp upstream region from the transcription start site in PlantCARE (http://bioinformatics.psb.ugent.be/webtools/plantcare/html/, (accessed on 1 March 2022)). In total, we screened 94 cis-acting regulatory elements in the promoter sequences of 79 *ZmPUB* genes (Appendix A, Figure 6). This included 3 core promoter elements, 19 hormone-responsive elements, 46 biotic- and abiotic-stress-response elements, 13 metabolism- and development-related elements, and 13 unknown function elements. The core promoter elements (CAAT-box, TATA-box) were present in all *ZmPUB* genes. The hormone-response elements consisted of MeJA and JA (4), ABA (4), ETH (3), GA (3), IAA (3), and SA (2). There were around 2–12 hormone response elements identified in each *ZmPUB* gene. Among these hormone-response elements, the MeJA and JA response element (as-1 and CGTCA-motif), ETH response element (MYC), and ABA response element (ABRE) were present in more than 80 percent of *ZmPUB* genes. Moreover, 46 distinct elements contributed to the biotic and abiotic stress responses, which included pathogen defense (3), light (28), drought (5), wound (3), cold (2), hypoxia (2), heat (1), salt (1), low osmotic pressure (1), and other abiotic stress (1). Among these, the light-response element (G-box) was present in the promoter regions of 84.4% *ZmPUB* genes. Additionally, the metabolism- and development-related elements harbored five seed or endosperm development elements (O^2^-site, GCN4_motif, OCT, RY-element, and MBSI), three meristem expression-associated elements (CAT-box, CCGTCC-box, and HD-Zip 1), two gene expression-related elements (CARE and AT-rich element), one circadian regulatory element (circadian), one root-specific regulatory element (motif I), and one cell cycle regulatory element (MSA-like).

### 2.5. Expression Profiles of ZmPUB Genes during Root Development

To explore a possible role of *ZmPUB* genes during root development, we employed the RNA-seq data from the MaizeGDB website, which is a submission by prior study [37]. The RNA-seq data revealed that *ZmPUB* genes were differentially expressed during root development (Figure 7, Appendix A). Three days after seed sowing, 21 *ZmPUBs* were upregulated in the differentiation zone of the primary root, whereas, only eight and four *ZmPUBs* were upregulated in the meristematic zone and stele of the primary root, respectively. Further, 6–7 days after seed sowing, 19 and 18 *ZmPUBs* exhibited elevated expression in the primary and seminal roots. In the V7 stage, 31, 21, and 2 *ZmPUBs* were highly expressed in the crown root nodes 1 through 3, crown root node 4, and crown root node 5, respectively. In the V13 stage, a total of 26 *ZmPUBs* revealed elevated expression in the crown root node 5. Interestingly, we noticed that *ZmPUB4*, *ZmPUB8*, *ZmPUB16*, *ZmPUB23*, *ZmPUB27*, *ZmPUB35*, *ZmPUB43*, *ZmPUB49*, *ZmPUB66*, and *ZmPUB78* showed a relatively high expression in varying root locations (FPKM > 10) (Appendix A). Among these genes, four *ZmPUB* genes contained the GA response element, five *ZmPUB* genes contained the IAA response element, and nine *ZmPUB* genes contained the meristem expression-related element (Appendix A). Lastly, seven *ZmPUB* genes were not at all expressed in the root.

### 2.6. Expression Profiles of ZmPUB Genes during Leaf Development

To explore the function of *ZmPUBs*, we analyzed the expression profiles of *ZmPUBs* during leaf development, and the gene expression data were provided by a previous study [38]. In the previous study, the fully expanded third leaf of the seeding maize B73 plants was sliced into 15 sections (M1–M15) from the base to the tip, and a gradient gene expression analysis was performed from immature to mature leaf tissue. There were 34 highly expressed *ZmPUB* genes, which were mainly concentrated within the basal part of the leaf (Figure 8, Appendix A). These evidences suggested that these genes may contribute to cell division and elongation. Eight *ZmPUB* genes were identified, which were highly expressed only in the middle region of the leaf. Three *ZmPUB* genes, namely *ZmPUB1*, *ZmPUB48*, and *ZmPUB62*, were only upregulated in the tip of the leaf. In addition, we identified *ZmPUB43*, *ZmPUB44*, *ZmPUB53*, *ZmPUB54*, *ZmPUB62*, and *ZmPUB78* with very high expression (FPKM > 20) in the different parts of the leaf, indicating that these may significantly modulate leaf development. The promoter regions of the highly expressed genes were further analyzed; six genes contained the MeJA, JA, ETH and light-response elements, five genes contained the ABA response elements, five genes contained the meristem expression-related element, and four genes contained the GA response elements (Appendix A). Therefore, it is our belief that these highly expressed genes in various locations of leaves may be crucial for photosynthesis, differentiation, and hormonal responses. Lastly, we also found that seven *ZmPUB* genes were not at all expressed in the leaves.

### 2.7. Expression Profiles of ZmPUB Genes during Seed Development

To explore the function of *ZmPUB* genes during seed development, we gathered the seed development RNA-seq data from prior investigations (Appendix A) [39,40]. Overall, 50 *ZmPUB* genes were expressed during the whole-seed development. Among them, 31 *ZmPUB* genes were present within the embryo, 9 *ZmPUB* genes were present in the endosperm, and 12 *ZmPUB* genes were present in both the endosperm and during embryo development (Figure 9). We also observed multiple highly expressed *PUB* genes, including *ZmPUB3*, *ZmPUB4*, *ZmPUB8*, *ZmPUB23*, *ZmPUB24*, *ZmPUB43*, and *ZmPUB52* (FPKM > 10), which suggested that they may associate with seed development (Appendix A). Among these genes, seven genes harbored the ETH and ABA response elements, six genes contained the meristem expression-related element, four genes contained the gliadin metabolic regulatory element, three genes contained the MeJA and JA response elements, two genes contained the GA response elements, one gene contained the IAA response elements, and one gene contained the endosperm formation regulatory element (Appendix A). In previous studies, hormones were reported as information transfer substances, which served essential functions in crop development, particularly, during grain filling [41,42,43]. This indicates that these highly expressed genes may participate in seed development. Additionally, we also identified 10 *ZmPUB* genes that were not expressed during seed development.

### 2.8. Expression Profiles of ZmPUB Genes during Abiotic Stress

To elucidate changes in the transcriptome response to abiotic stress, RNA-seq was carried out in maize under control and stress conditions [44]. Under cold stress (7 °C for 16 h), 31 *ZmPUB* genes were markedly elevated in maize seedlings relative to control, 12 of which showed high expression (fold change >1). *ZmPUB45* was the most up-regulated gene (fold change >36). Twenty-four *ZmPUB* genes were down-regulated in maize seedlings under cold stress. Among them, *ZmPUB29*, *ZmPUB61*, and *ZmPUB73* displayed the most down-regulated expression, which were 31%, 24%, and 35% of control, respectively (Figure 10, Appendix A). Under heat stress (50 °C for 4 h), sixteen *ZmPUB* genes were highly expressed in maize seedlings relative to control, seven of which showed extremely high expression (fold change >1). *ZmPUB30* showed the highest up-regulated expression, which was up-regulated by 12.5 times. Forty *ZmPUB* genes were down-regulated, nineteen of which were significantly down-regulated by more than 50% (Figure 10, Appendix A). Under salt stress, twenty-six *ZmPUB* genes were up-regulated in maize seedlings, and among them, eight genes were up-regulated by more than 1-fold, and *ZmPUB45* and *ZmPUB57* were the most up-regulated genes, with increases by 25.5 and 84.8 times, respectively. Thirty-two genes were down-regulated in maize seedlings as compared to controls, and four genes (including *ZmPUB1*, *ZmPUB27*, *ZmPUB40*, and *ZmPUB78*) showed the most down-regulated expression, with reductions of 67%, 69%, 82%, and 63%, respectively (Figure 10, Appendix A). Under UV stress, thirty-nine *ZmPUB* genes were up-regulated, and seven genes were up-regulated more than 1-fold; among them, *ZmPUB27* and *ZmPUB45* had the most up-regulated expression, which were 5.18 and 19.8 times, respectively. Twenty-five *ZmPUB* genes were down-regulated, four of which were down-regulated by more than 50% (Figure 10, Appendix A). In addition, nine *ZmPUB* genes, including *ZmPUB7*, *ZmPUB9*, *ZmPUB12*, *ZmPUB28*, *ZmPUB41*, *ZmPUB56*, *ZmPUB74*, *ZmPUB77*, and *ZmPUB79*, were not influenced by various abiotic stressors.

Drought stress affected plants particularly during reproduction. To investigate the potential role of *ZmPUB* genes in drought stress, we collected the RNA-seq data of drought-exposed and appropriately watered fertilized ovary and basal leaf meristem tissue (Figure 10, Appendix A) [45]. Under drought stress, 36 up-regulated and 30 down-regulated genes were screened in the ovary, among which 41 genes were altered more than 50%. *ZmPUB49* was the gene with the maximum change in expression, which was up-regulated by 6.5 times. There were 32 up-regulated and 30 down-regulated genes in the basal leaf meristem following drought stress, among which 20 genes were altered more than 50%. Four genes (including *ZmPUB2*, *ZmPUB31*, *ZmPUB57*, and *ZmPUB78*) exhibited the maximum alterations in expression, which increased by 14.0, 5.3, 5.6, and 8.0 times, respectively. Lastly, nine genes were not expressed under drought stress in the ovary or basal leaf meristem tissue.

### 2.9. Expression Profiles of ZmPUB Genes during Hormonal Treatment

To further elucidate whether hormone affect *ZmPUBs* expression, we conducted RNA-seq under the hormone treatments (ABA, IAA, and GA). Based on our observations, 42 (53.16%) *ZmPUB* genes were detected in the leaf tissue (FPKM > 1) (Figure 11, Appendix A). Thirty-nine *ZmPUB* genes responded to the hormones ABA, IAA, and GA but with variable degrees of sensitivity. Under ABA treatment, 35 *ZmPUB* genes were significantly altered in expression pattern, and among them, 13 genes were down-regulated, 20 genes were up-regulated, and 2 genes were down-regulated at the early stage (3 or 6 h) and up-regulated at the late stage (12 h). Under IAA treatment, 15 *ZmPUB* genes were altered, among which 5 were reduced, and 10 were augmented. Under GA treatment, 17 *ZmPUB* genes were altered, 6 of which were significantly diminished at one stage, and 11 were significantly elevated at one or two stage. As a result, we identified eight *ZmPUB* genes that showed the highest degree of hormone-induced expression, and they were *ZmPUB1*, *ZmPUB4*, *ZmPUB43*, *ZmPUB45*, *ZmPUB57*, *ZmPUB64*, *ZmPUB68*, and *ZmPUB78*, respectively. These evidences suggested that these *ZmPUB* genes contributed to the hormone-responsive network.

## 3. Discussion

The *PUB* genes are common among mammals, plants, and microorganisms, and they encode the PUB ligase enzyme, a key enzyme in ubiquitin proteasome-mediated degradation [8]. The genome-wide analysis of *PUB* genes was performed in numerous species, and they were found to play essential roles in modulating plant development and response to biotic, abiotic, and hormonal stressors [1,11,22,23]. Herein, we screened 79 *ZmPUB* genes in the maize genome and designated *ZmPUB1* through *ZmPUB79* based on their location on the chromosome.

Phylogenetic analysis revealed that these *ZmPUB* genes can be clustered into seven categories. Similar results were found in rice, soybean, barley, citrus, and banana (*Musa paradisiaca*), wherein the genes were classified into 6–8 groups using phylogenetic analysis [15,16,18,46]. Further analysis revealed that the 36 *ZmPUB* genes encoded proteins that harbor only the U-box domain, and 43 *ZmPUB* genes encoded proteins that contain the U-box domain, along with additional 10 domains, including the ARM repeat (25), coil coil region(10), pkinase-tyr (9), TPR (2), UFD2P_CORE (2), ZnF_TTF (1), WD40 repeat (1), KAP (1), RS4NT (1), and USP (1) domains (Appendix A). In *Arabidopsis*, 16 AtPUB proteins harbor only the U-box domain, whereas 48 AtPUB proteins contain both the U-box domain and several other domains, such as ARM repeat (27), kinase (14), WD40 repeat (2), MIF4G (2), cyclophilin (1), UFD2 (1), and TPR domains (1) [13]. In rice, the domain composition of PUB proteins are similar to Arabidopsis; all OsPUB proteins contain the U-box domain, and 57 OsPUB proteins contain six types of domains except MIF4G [12]. In higher eukaryotes, genomic evolution can lead to the gene expansion and functional differentiation of gene families. Being the core domain of the *PUB* gene family, the U-box domain is evolutionarily conserved. Meanwhile, there are plenty of other domains observed in numerous species, which might introduce functional diversity to the *PUB* genes.

In eukaryotes, genes are usually composed of exons, introns and untranslated regions (UTRs) at the 5′ and 3′ ends. The structural organization is closely related to gene function and family evolution [47]. In the maize *PUB* gene family, there was a wide variation in the number of exons; the mean number of exons per gene was 3.61, and it ranged from 1 to 16. Around 37.97% of the *ZmPUB* genes contained only one exon and no intron (Figure 2, Appendix A). In prior investigations, about 1 to 17 exons were identified in the *PUB* genes of tomato and *Medicago*, with an average of four exons per gene, and the *PUB* genes with only one exon and no intron accounted for 34.14% and 40.32% of all *PUB* genes [17,21]. The structural organization of *PUB* genes are similar in different species. It is suggested that the *PUB* gene family is evolutionarily conserved. Exons possess the core information needed by the cell to synthesize protein, whereas introns protect the coding proteins from randomly generating deleterious mutations [48]. In the *PUB* gene family, numerous intron-less genes were reported in multiple species, such as tomato, grape, citrus, Medicago, and Chinese cabbage [14,17,18,21,49], suggesting a strong structural integrity of *PUB* genes. Furthermore, the conserved motif distributions suggested a strong structural and functional similarity among maize *PUB* genes (Figure 2, Appendix A). For instance, motifs 1, 3, and 4 are highly conserved and exist in almost all *PUB* genes. We also observed some unique features in some groups of the maize *PUB* gene family. Motifs 5 and 2 tandem sequences were mainly located in group IV. Motifs 6, 7, and 9 were the exclusive feature of group II. These analyses are highly beneficial to scientists who are interested in the evolution, structure, and function of the *PUB* gene family.

Gene duplication (GD), mutation, and natural selection are the major sources of new genes and functions, and they provided a basis for biodiversity [50]. Gene duplication and syntenic analysis identified 25 pairs (43 *ZmPUB* genes) of SD and one pair of TD (*ZmPUB43/ZmPUB44*) in the maize *PUB* gene family (Table 1). GD is speculated to be the major contributor of expansion and diversity in the maize *PUB* gene family. We next evaluated the domain composition of the maize duplicate genes. New domains were introduced to the newly duplicated genes; five genes were added to the ARM domain (*ZmPUB10*, *ZmPUB19*, *ZmPUB39*, *ZmPUB43*, and *ZmPUB63*), two genes to the coil coil region (*ZmPUB62* and *ZmPUB8*), and one gene to the KAP domain (*ZmPUB49*) (Table 1 and Appendix A). The addition of new domains to the duplicated genes may have introduced diversification of function to *PUB* genes. The maize *PUB* gene family was predicted to harbor GD events around 13–84 Mya and 129–215 Mya (Table 1). Based on our analysis, there were approximately 0.05–0.99 synonymous substitutions per site, which indicated that the maize *PUB* gene family evolved under purifying selection, whereby deleterious mutations were eradicated while the core functional domains were conserved [51]. In addition, the maize *PUB* genes displayed a stronger association with rice than Arabidopsis (Figure 5). This suggested that rice is a more suitable reference for the functional study of the maize *PUB* gene family.

Cis-acting regulatory element analysis identified 94 kinds of elements in the promoter sequences of the maize *PUB* gene family. These were 19 hormone-response elements, 46 biotic- and abiotic-response elements, 13 metabolism- and development-related elements, and 13 unknown function elements (Appendix A). Similar element patterns were reported in tomato, citrus, and Medicago, which suggested a common relationship between *PUB* genes and stresses as well as hormonal and developmental modulatory mechanisms in plants [17,18,52].

Emerging evidences revealed that the *PUB* genes played essential functions in modulating plant development and in developing tolerance to abiotic stresses such as cold, heat, salt, and drought [53,54,55,56]. Using available transcriptome data and RNA-seq analysis under various treatments, we examined the expression profile of the maize *PUB* genes in this research. Our work could provide new clues to the biological function of *PUB* genes in maize.

The *PUB* genes were known to regulate development of roots, stems, leaves, flowers, and fruits in plants [17,57]. Herein, we identified different expression profiles of maize *PUB* genes during the root development (Figure 7, Appendix A). In total, 10 genes (*ZmPUB4, ZmPUB8*, *ZmPUB16*, *ZmPUB23*, *ZmPUB27*, *ZmPUB35*, *ZmPUB43*, *ZmPUB49*, *ZmPUB66*, and *ZmPUB78*) were highly expressed at certain stages of root development (FPKM > 10). Further analysis revealed that GA, IAA, and meristem expression-related elements were present in the promoter of these genes (Appendix A). The root apical meristem-specific genes played essential roles in root development during early somatic embryogenesis [58]. IAA strongly promoted the development of adventitious roots, lateral roots, root hairs, and primary roots. GA determined cell growth polarity in the root cortex of maize [59,60,61]. It is suggested that these genes participate in root morphogenesis. Six highly expressed genes were observed during leaf development, and they were *ZmPUB43*, *ZmPUB44*, *ZmPUB53*, *ZmPUB54*, *ZmPUB62*, and *ZmPUB78*, respectively (Figure 8, Appendix A). We screened the MeJA, JA, ETH, and light-response elements in the promoters of these genes (Appendix A). MeJA, JA, and ETH regulated plant growth. The enhanced endogenous MeJA levels induced morphological alteration within leaves in the transgenic soybean plants [62]. JA and ETH may positively regulate cotton leaf senescence [63]. Leaf is an important photosynthetic organ in plants, and light is known to modulate plant morphogenesis and the leaf ultrastructure [64]. The presence of MeJA, JA, ETH, and light-response elements provided evidence that these genes may be involved in leaf development. Similarly, seven highly expressed genes (*ZmPUB3*, *ZmPUB4*, *ZmPUB8*, *ZmPUB23*, *ZmPUB24*, *ZmPUB43*, and *ZmPUB52*) were spotted during seed development, four of which harbored the gliadin metabolic regulatory element (O^2^-site) in their promoter regions (Figure 9, Appendix A). Gliadin is the most abundant storage protein in maize seed, and it is critical for the nutritional profile of maize seeds [65]. Based on our gene expression analysis, these genes were ubiquitously expressed at 2–14 days of whole-seed development. This suggested that they may participate in seed development by modulating protein accumulation within the endosperm.

The *PUB* genes modulate response to abiotic and hormonal stresses [52,58]. Based on the RNA-Seq data, there were several stress- and hormone-responsive genes, such as *ZmPUB1*, *ZmPUB27*, *ZmPUB30*, *ZmPUB31*, *ZmPUB43*, *ZmPUB45*, *ZmPUB57*, and *ZmPUB78* (Appendix A).

*ZmPUB1* and *OsPUB75* were homologous genes. *OsPUB75* encoded a cytosolic RING-type E3 ubiquitin ligase, which was a crucial negative modulator of abiotic stress. Under salinity and mannitol stress, *OsPUB75* was transcriptionally repressed in Arabidopsis [66]. In maize, *ZmPUB1* displayed an expression profile similar to *OsPUB75*, and it was down-regulated by 26–83% under heat, salt, UV, drought, and ABA treatments (Appendix A). This suggested that *ZmPUB1* may participate in the negative regulation of abiotic stresses and hormonal responses. *ZmPUB27* was homologous with *AtPUB16* and *OsPUB3*. In Arabidopsis, *AtPUB16* participated in the GA pathway, which favored self-pollination [67]. In maize seedling, a similar *ZmPUB27* up-regulation was observed at 3 and 12 h after GA treatment. This indicated that *ZmPUB27* may contribute to the GA response. In rice, *OsPUB3* was a positive regulator of cold stress, and its over-expression in transgenic plants showed enhanced tolerance to cold stress compared to wild-type plants [53]. *ZmPUB27* expression was elevated under conditions of cold, drought, and UV, and it was reduced after heat and salt treatment. This suggested that *ZmPUB27* was involved in abiotic stresses. *ZmPUB30* and *ZmPUB57* were a pair of duplicated genes, and they were homologous to *OsPUB5*, *AtPUB18*, and *AtPUB19*. In rice, *OsPUB5* encoded a copper (Cu) transporter 6 protein, which was highly expressed in the node, and it blocked the cadmium (Cd) upward transport [68]. *AtPUB18* and *AtPUB19* were positively regulated by ABA and salt, and their double mutants could not respond well to ABA and the salt-based suppression of seed germination relative to wild-type plants [69]. Under cold, heat, salt, and ABA treatments, *ZmPUB30* and *ZmPUB57* were up-regulated by 0.5–12.5 or 3.67–84.82 times. The conserved motif analysis revealed that *ZmPUB57* contained more of motifs 2, 5, and 8 than *ZmPUB30*. This may greatly enhance the stress response ability of *ZmPUB57*. *ZmPUB31* and *ZmPUB45* were a pair of duplicated genes. Under cold, heat, salt, drought, UV, and ABA stresses, *ZmPUB31* and *ZmPUB45* were up-regulated by 0.19–3.8 and 3.1–36.6, respectively (Appendix A). MeJA and JA were naturally occurring physiologically active materials that respond to exoteric stimulations by transmitting stress signals and activating the stress-resistant genes in plants [70,71,72]. Compared to *ZmPUB31*, there were more JA and MeJA response elements in the promoter sequence of *ZmPUB45* (Appendix A). This may enhance the responsive capacity of *ZmPUB45* under environmental stress. *OsPUB45* was homologous with *ZmPUB31* and *ZmPUB45*. In rice, *OsPUB45* was up-regulated many-folds under salt, drought, and cold stresses [73]. We observed a similar expression pattern between *ZmPUB45* and *OsPUB45*, which indicated that these two genes may possess comparable functions. *ZmPUB43* was the homologous gene of *OsPUB8*. In rice, *OsPUB8* was markedly up-regulated during drought stress [74]. *ZmPUB43* was up-regulated by 29–36% in the maize ovary and meristem under drought stress (Appendix A). Further analysis revealed that there were significant alterations in expression of *ZmPUB43* after salt, UV, and ABA treatments. ABA was critical for multiple biological processes, such as seed dormancy, germination, and adaptive responses to abiotic stresses [75,76]. This suggested that *ZmPUB43* may contribute to abiotic stress responses via the ABA pathway. *ZmPUB78* was homologous to *AtPUB11*. *AtPUB11* was up-regulated under drought and ABA treatments, and its silent mutant was highly tolerant to drought stress compared to the wild type [77]. In maize, the expression levels of *ZmPUB78* were up-regulated by 2.8–8.0 fold after ABA and drought treatments, which suggested that *ZmPUB78* may negatively regulate the ABA-mediated drought response. In contrast, *ZmPUB78* was down-regulated by the cold, heat, and salt stresses, thus indicating that *ZmPUB78* may also be involved in muti-abiotic stresses. Above all, these results provided insights into the possible biological functions of the maize *PUB* gene family. Our extensive analyses aided in the selection of specific *PUB* genes for additional functional research so as to improve the genetic agronomic characteristics and strengthen environmental resistance within maize.

## 4. Materials and Methods

### 4.1. Screening for PUB Genes in the Maize Genome

The maize genome sequences (B73 RefGen_v4) were obtained from the Ensembl Plants database (https://plants.ensembl.org/index.html (accessed on 1 March 2022)). The Hidden Markov Model (HMM) profile of the U-box domain (PF04564) was retrieved from the Pfam database (http://pfam.xfam.org/ (accessed on 5 March 2022)). The HMMER program was searched in the maize genome with default parameters and a significant e^−3^ value. Next, we confirmed the putative *ZmPUB* genes using the Pfam (http://pfam.xfam.org/ (accessed on 5 March 2022)), SMART (http://smart.embl-heidelberg.de/ (accessed on 5 March 2022)), and NCBI CDD (https://www.ncbi.nlm.nih.gov/cdd (accessed on 5 March 2022)) databases. The putative ZmPUB protein MW (molecular weight) and PI (isoelectric point) were analyzed using ExPASy (http://www.expasy.org (accessed on 5 March 2022)). WoLF PSORT (https://www.genscript.com/wolf-psort.html (accessed on 5 March 2022)) was employed to estimate the subcellular localization of ZmPUB proteins.

### 4.2. Multiple Alignments and Phylogenetic Analyses

Multiple sequence alignments were performed using the U-box domain sequences of the 219 PUB proteins (79 *ZmPUBs*, 77 *OsPUBs*, and 64 *AtPUBs*) using MEGA 7 [78]. A phylogenetic tree was generated using the neighbor-joining (NJ) method and the following parameters: Poisson correction, complete deletion, and 1000 bootstrap replicates, and visualization was done in the EvolView v2 software [79,80]. The *Oryza sativa* and *Arabidopsis thaliana* PUB protein sequences were obtained from the NCBI website (https://www.ncbi.nlm.nih.gov/ (accessed on 6 May 2022)), as reported in previous studies [12,13].

### 4.3. Genetic Structure and Conserved Motif Analyses

The genetic structure analysis was conducted via comparison of the CDS and DNA sequences of the *ZmPUB* genes in GSDS 2.0 (http://gsds.cbi.pku.edu.cn/ (accessed on 6 May 2022)), SMART (http://smart.embl-heidelberg.de/ (accessed on 6 May 2022)) and Pfam (http://pfam.xfam.org/ (accessed on 6 May 2022)) softwares [81]. The conserved motifs were analyzed in MEME using parameters as follows: maximum motif number (10) and motif length (6–100 amino acid residues) (http://meme-suite.org (accessed on 6 May 2022)) [82], and visualization was performed in Tbtools (https://github.com/CJ-Chen/Tbtools (accessed on 6 May 2022)).

### 4.4. Chromosomal Mapping, Gene Duplication (GD), and Synteny Analyses

The gene lengths and position data were acquired from the maize genome (B73 RefGen_v4). Next, MapChart was employed for the construction of the chromosomal localization map [83]. To assess GD events, the coding sequences of *PUB* genes from *Arabidopsis*, maize, and rice were aligned in BLASTp using an E-value < 1 × e^−10^ cut-off. MCScanX was employed to assess GD events and examine syntenic association between different species, and visualization was performed in Circos [84,85]. The Kaks_Calculator computed the nonsynonymous (Ka) and synonymous (Ks) substitution rates of duplicated gene pairs, and the approximate time of GD was estimated as follows: T = (Ks/2λ) × 10^−6^, where λ = 6.5 × 10^−9^ [86].

### 4.5. Cis-Acting Regulatory Element Analysis

We downloaded the 1500 bp upstream flanking sequences from the transcription start site of each *ZmPUB* gene. Then the cis-acting regulatory element analysis was carried out in PlantCARE (http://bioinformatics.psb.ugent.be/webtools/plantcare/html/ (accessed on 10 May 2022)) [87].

### 4.6. RNA-seq and Gene Expression Profile

The RNA-seq information for *ZmPUB* genes in numerous tissues, developmental stages, and stress conditions were acquired from NCBI (https://www.ncbi.nlm.nih.gov/ (accessed on 6 June 2022)) and MaizeGDB (https://www.maizegdb.org/ (accessed on 6 June 2022)) and were reported in previous studies [38,39,40,88]. The expression heat maps of *ZmPUB* genes were drawn in the omicshare website (https://www.omicshare.com/ (accessed on 10 June 2022)).

*Zea mays* cv. B73 was employed for the examination of gene profile in response to multiple hormonal treatments. All seeds were cultivated in commercial soil at 28 °C in a photoperiod of 16 h light/8 h dark. Three-leaf stage seedlings were treated with 100 μmol/L ABA, 100 μmol/L IAA, and 100 μmol/L GA in 0.1‰ Tween-20, respectively. The controls were treated with water in 0.1‰ Tween-20. The leaves were harvested at 0, 3, 6, 12, and 24 h after hormonal treatments. Three replicates of three plants were used in each treatment. All samples were flash-frozen in liquid nitrogen prior to storage in −80 °C until subsequent analysis.

Total RNA extraction employed Trizol (Invitrogen, Carlsbad, CA, USA). The quality of RNA was assessed via the Agilent 2100 Bioanalyzer (Agilent Technologies, Santa Clara, CA, USA). Transcripts were enrich-fragmented into small pieces, which were then converted to cDNA. The cDNA library was generated with the NEBNext Ultra RNA Library Prep Kit for Illumina (NEB #7530, New England Biolabs, Ipswich, MA, USA), and sequencing was completed with Illumina Novaseq 6000 by Gene Denovo Biotechnology Co., (Guangzhou, China). To quantify gene abundance, we filtered reads and mapped the clean reads to the reference genome with the fastp, HISAT2. 2.4, and StringTie v1.3.1 softwares [89,90,91]. Finally, the FPKM (fragment per kilobase of transcript per million mapped reads) gene values were computed in the RSEM software [92].

### 4.7. Statistical Analyses

To compare the variation of gene expression, SPSS 12.0 (SPSS Inc., Chicago, IL, USA) was employed to carry out the least significant difference (LSD) test. Data processing and visualization were done with GraphPad Prism 5. Individual data were presented as mean ± standard error (SE) of three experimental replicates.

## 5. Conclusions

Herein, we conducted an extensive analysis of *PUB* genes in maize. We identified 79 *ZmPUB* genes, which were stratified into 7 categories. Each group exhibited similar exon-intron structures and motif compositions. Using gene duplication and synteny analysis of *PUB* genes, we obtained important clues regarding the evolutionary profiles of maize *PUB* genes. *PUB* genes strongly regulated plant development as well as response to abiotic stresses and hormones. Our phylogenetic and gene expression analyses provided useful information for enhancing our comprehension of the biological roles of the *PUB* genes in maize.

## Figures and Tables

**Figure 1 plants-11-02459-f001:**
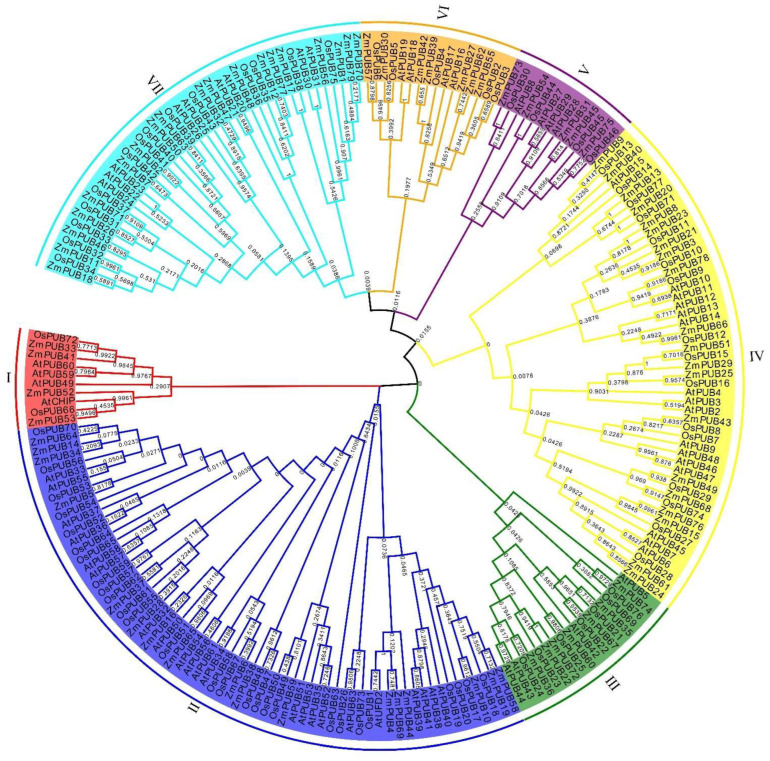
The phylogenetic tree of the *PUB* gene family. Multiple sequence alignments of the U-box domain sequences of rice, Arabidopsis, and maize were performed using MEGA 7. A phylogenetic tree was generated via the neighbor-joining technique with 1000 bootstrap values. The *PUB* genes were then separated into seven distinct categories marked by distinct colors.

**Figure 2 plants-11-02459-f002:**
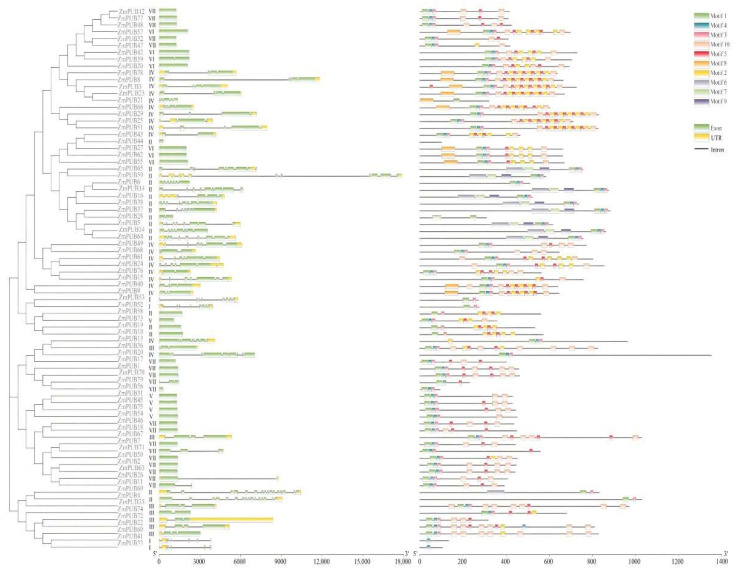
Gene structures and conserved motifs of the maize *PUB* gene family. A phylogenetic tree of *PUB* genes. Multiple sequence alignment of the U-box domain sequences in maize was carried out with Clustal W. The neighbor-joining (NJ) tree was generated via MEGA X with 1000 bootstrap replicates; The gene structure of *PUB* genes. The yellow boxes, green boxes, and black lines represent exon, intron, and UTR (untranslated region), respectively; The conserved motif of *PUB* genes. MEME analysis revealing the conserved motifs belonging to the maize *PUB* gene family. The colored boxes on the right denote 10 motifs.

**Figure 3 plants-11-02459-f003:**
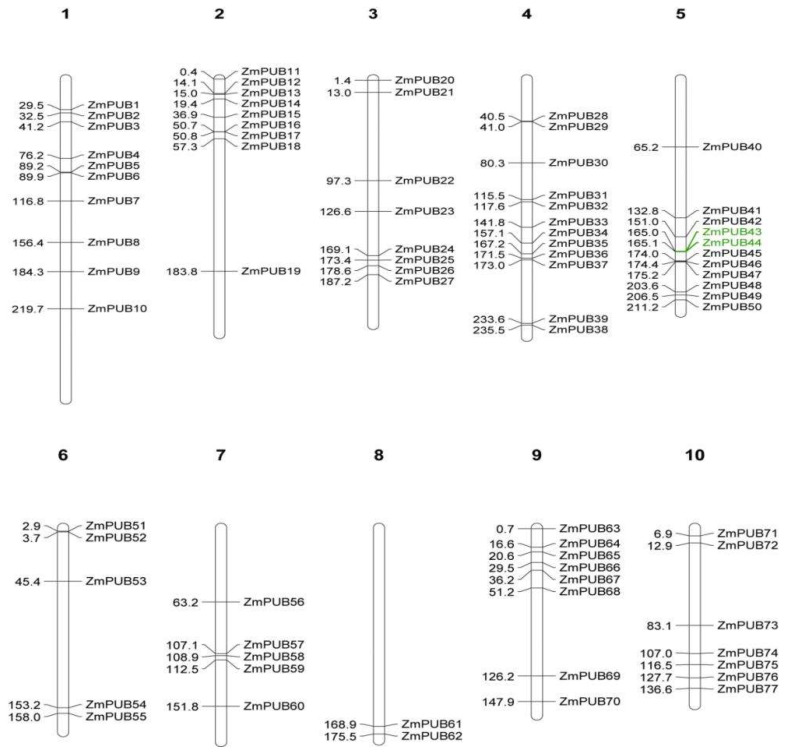
Chromosomal locations of *PUB* genes in *Zea mays* (Mb). A total of 77 *ZmPUB* genes were mapped to the 10 chromosomes of maize; the other two genes were identified on unassembled genomic contigs. White bars denote the chromosome. The numbers 1–10 indicate chromosomes 1–10. The names and locations of individual *ZmPUB* genes are indicated on each bar. The genes colored in green represented tandem duplication (TD).

**Figure 4 plants-11-02459-f004:**
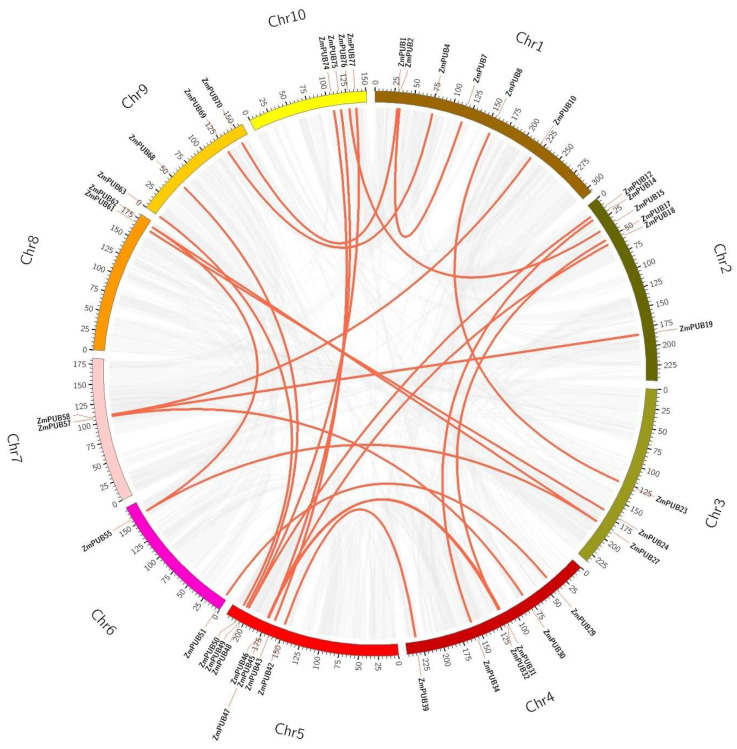
The Circos diagram of the *PUB* gene family in maize. The red lines represent the duplicated *ZmPUB* gene pairs. The gray lines represent the gene collinearity regions of the maize genome. The color bars represent the maize chromosome. The scale bars on chromosomes indicate chromosomal lengths (Mb).

**Figure 5 plants-11-02459-f005:**
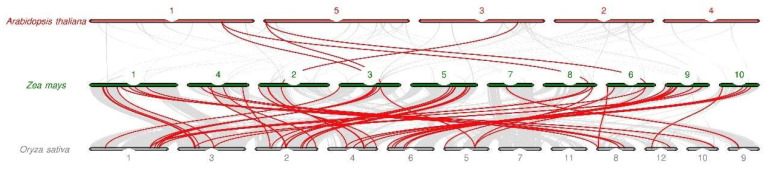
Synteny analysis of the *PUB* gene family. The background gray lines represent the collinear blocks within the genomes of Arabidopsis, maize, and rice. The red lines represent the collinear *PUB* gene pairs in Arabidopsis, maize, and rice.

**Figure 6 plants-11-02459-f006:**
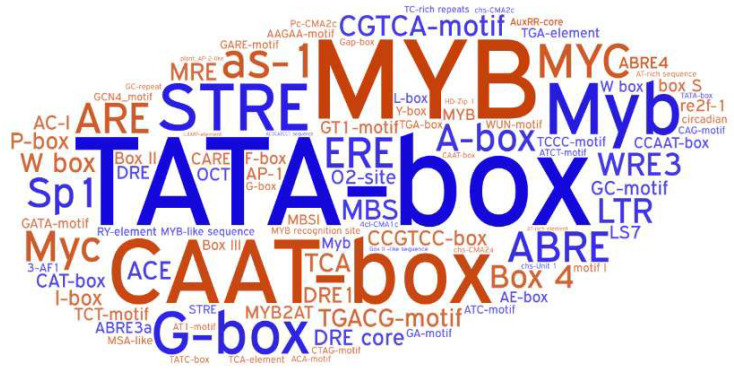
A word cloud image of the cis-acting regulatory element of 79 *PUB* genes in maize. The size and intensity indicate the frequency and occurrence.

**Figure 7 plants-11-02459-f007:**
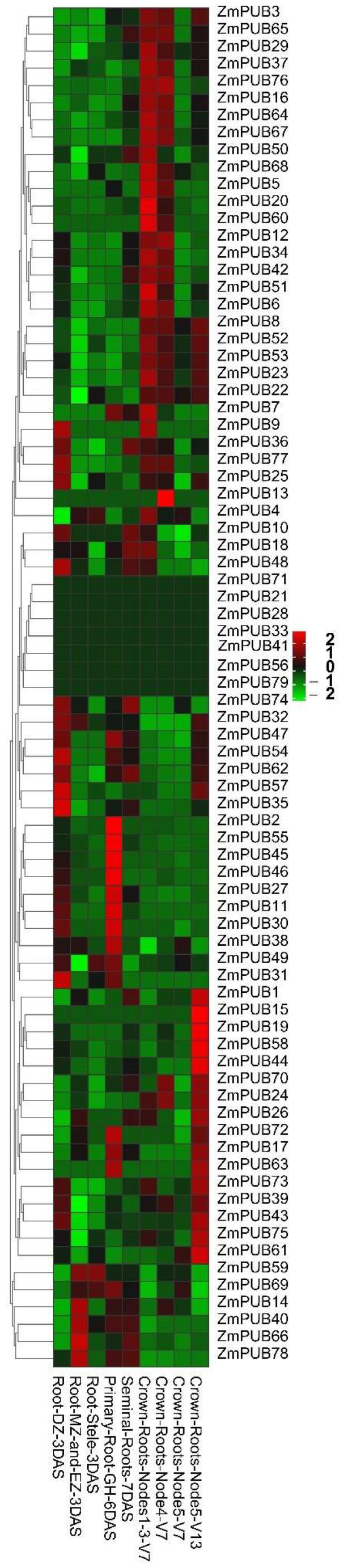
Expression profiles of *ZmPUB* genes during root development. DAS, the days after sowing; Vn, vegetative stage referencing the quantity of emerged leaves; Root_DZ_3DAS, primary root differentiation zone after sowing (3DAS); Root_MZ_and_EZ_3DAS, primary root meristematic zone after sowing (3DAS); Root_Stele_3DAS, primary root stele after sowing (3DAS); Primary_Root_GH_6DAS, primary root after sowing (6DAS); Seminal_Roots_7DAS, seminal root after sowing (7DAS); Crown_Roots_Nodes13_V7, crown roots nodes 1 through 3 in V7; Crown_Roots_Node4_V7, crown roots node 4 in V7; Crown_Roots_Node5_V7, crown roots node 5 in V7; Crown_Roots_Node5_V13, crown roots node 5 in V13. The bold represents highly expressed *ZmPUB* genes. The original expressions of individual *ZmPUB* genes are summarized in Appendix A.

**Figure 8 plants-11-02459-f008:**
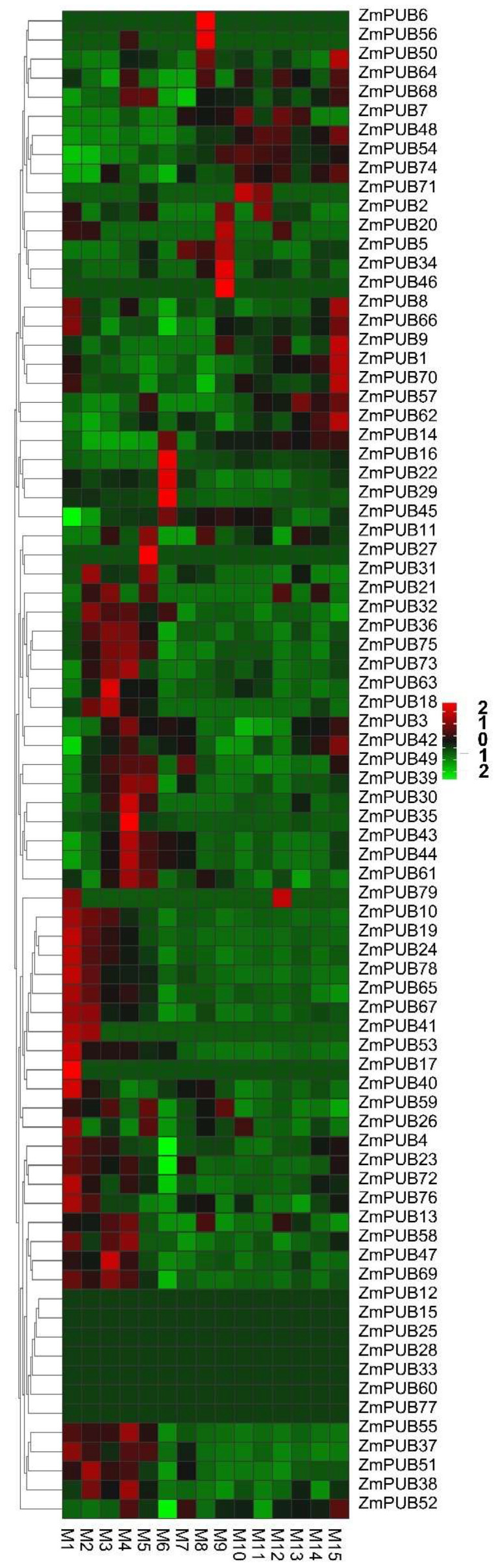
Expression profiles of *ZmPUB* genes during leaf development. The fully expanded third leaf of seeding maize B73 plants was sliced into 15 sections from the base to the tip, and they were designated as M1–M15. The original expression of *ZmPUB* genes are summarized in Appendix A.

**Figure 9 plants-11-02459-f009:**
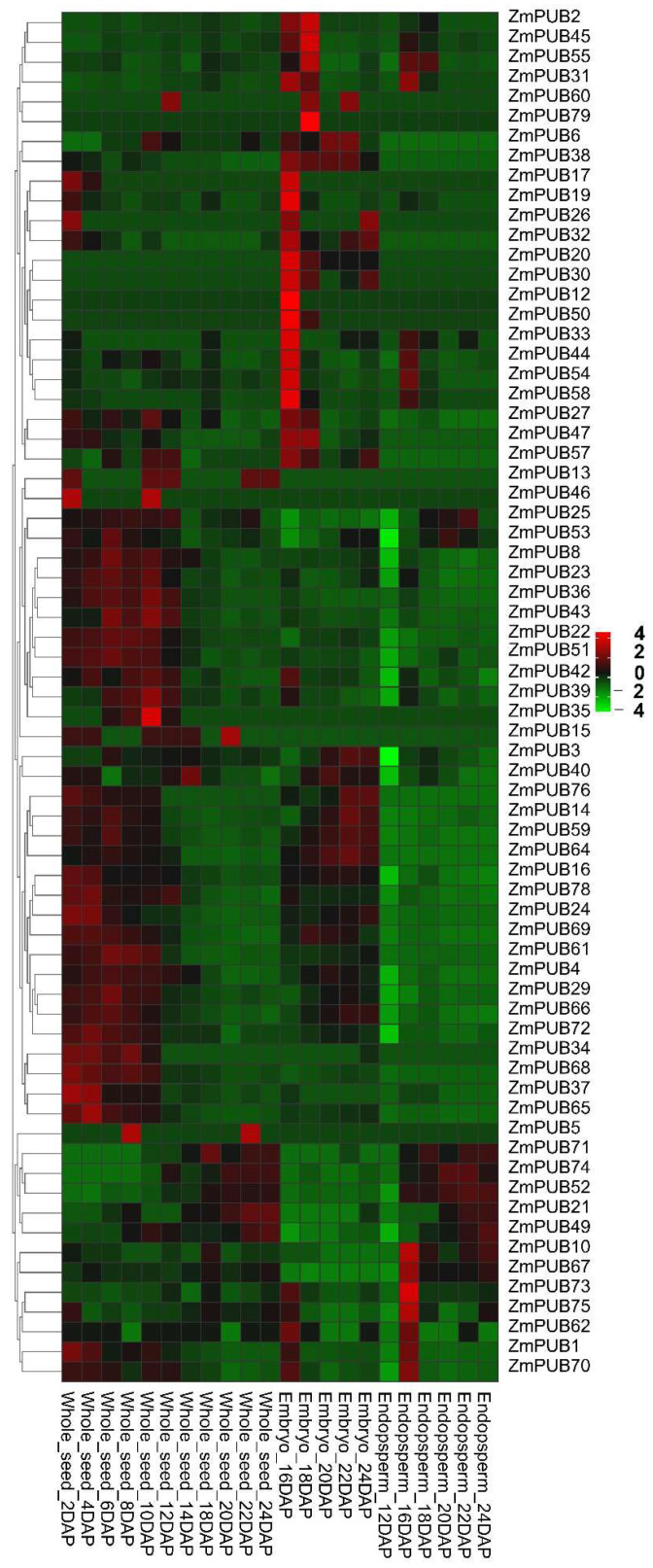
Expression profiles of *ZmPUB* genes during seed development. DAP, days after pollination. The original expression of individual *ZmPUB* genes are summarized in Appendix A.

**Figure 10 plants-11-02459-f010:**
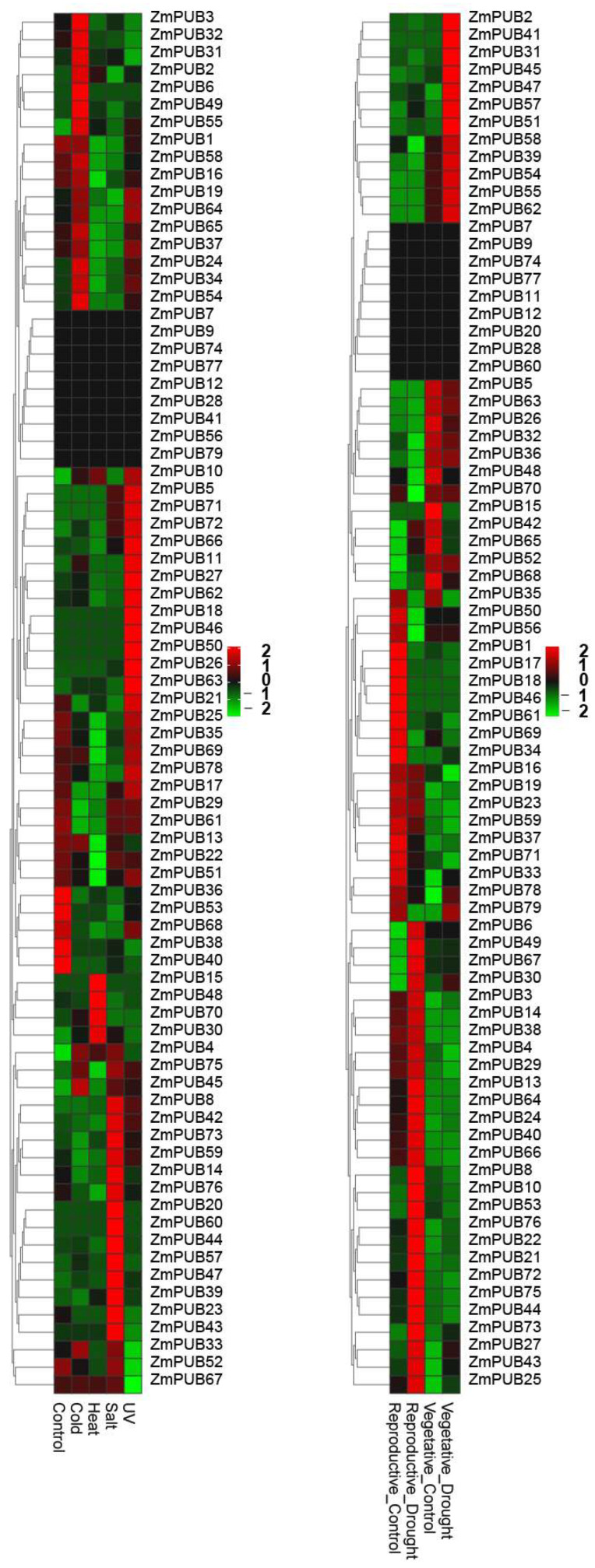
Expression profile of the *ZmPUB* genes under abiotic stresses. Reproductive_Control represents the well-watered fertilized ovary. Reproductive_Drought represents the drought-treated ovary. Vegetable_Control represents the well-watered fertilized basal leaf meristem tissue. Vegetable_Drought represents the drought-treated basal leaf meristem tissue. The original expressions of individual *ZmPUB* genes are summarized in Appendix A.

**Figure 11 plants-11-02459-f011:**
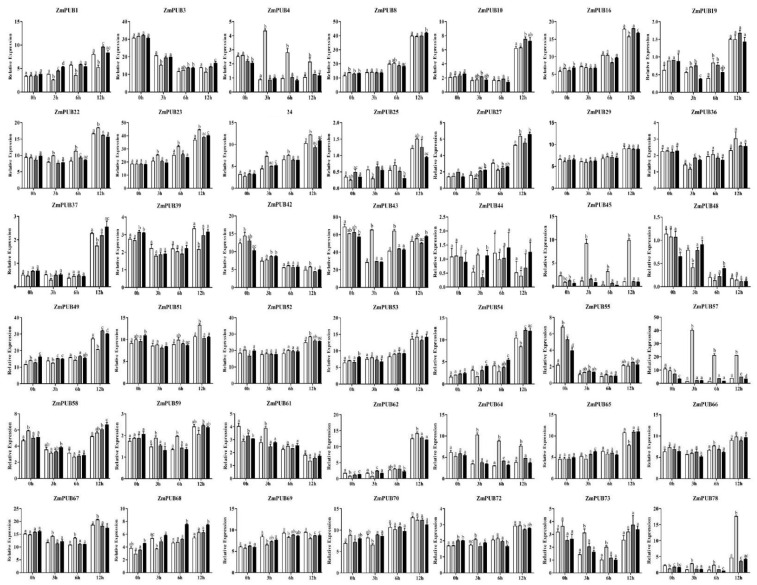
Expression profile of *ZmPUB* genes following hormone treatment. The white column represents control. The light gray column represents the ABA treatment. The dark gray column represents the IAA treatment. The black column represents the GA treatment. The level of significance (*p* < 0.05) among the different treatments is indicated by different letters. The original expressions of individual *ZmPUB* genes are summarized in Appendix A.

**Table 1 plants-11-02459-t001:** The segmental and tandem duplications of *ZmPUB* gene in maize.

Duplicated Gene Pairs	Ka	Ks	Ka/Ks	Duplicated Type	Purify Selection	Time (Mya)
*ZmPUB43/ZmPUB44*	0	0	NA	Tandem	NA	NA
*ZmPUB29/ZmPUB51*	0.04	0.17	0.25	Segmental	Purify selection	13.07
*ZmPUB55/ZmPUB27*	0.13	1.95	0.07	Segmental	Purify selection	150.27
*ZmPUB55/ZmPUB62*	0.13	1.69	0.07	Segmental	Purify selection	129.99
*ZmPUB4/ZmPUB69*	0.02	0.2	0.09	Segmental	Purify selection	15.01
*ZmPUB34/ZmPUB14*	0.99	1.03	0.96	Segmental	Purify selection	79.21
*ZmPUB63/ZmPUB50*	0.2	2.18	0.09	Segmental	Purify selection	167.62
*ZmPUB75/ZmPUB45*	0.17	2.81	0.06	Segmental	Purify selection	215.77
*ZmPUB31/ZmPUB45*	0.05	0.63	0.07	Segmental	Purify selection	48.32
*ZmPUB8/ZmPUB23*	0.06	0.46	0.12	Segmental	Purify selection	35.08
*ZmPUB27/ZmPUB62*	0.02	0.32	0.07	Segmental	Purify selection	24.28
*ZmPUB57/ZmPUB30*	0.22	2.66	0.08	Segmental	Purify selection	204.83
*ZmPUB49/ZmPUB68*	0.99	1.05	0.93	Segmental	Purify selection	81.07
*ZmPUB1/ZmPUB70*	0.02	0.32	0.05	Segmental	Purify selection	24.67
*ZmPUB32/ZmPUB17*	0.99	1.02	0.97	Segmental	Purify selection	78.39
*ZmPUB58/ZmPUB10*	0	0	NA	Segmental	NA	NA
*ZmPUB58/ZmPUB19*	1	1.01	0.99	Segmental	Purify selection	77.35
*ZmPUB46/ZmPUB18*	0.23	2.41	0.1	Segmental	Purify selection	185.29
*ZmPUB77/ZmPUB48*	0.18	2.37	0.08	Segmental	Purify selection	182.21
*ZmPUB32/ZmPUB47*	0.03	0.37	0.09	Segmental	Purify selection	28.78
*ZmPUB2/ZmPUB7*	0.33	2.04	0.16	Segmental	Purify selection	156.71
*ZmPUB61/ZmPUB24*	0.05	0.17	0.27	Segmental	Purify selection	13.26
*ZmPUB43/ZmPUB74*	0.99	1.04	0.95	Segmental	Purify selection	79.82
*ZmPUB42/ZmPUB39*	0.03	0.18	0.15	Segmental	Purify selection	13.78
*ZmPUB12/ZmPUB48*	0.18	2.03	0.09	Segmental	Purify selection	156.32
*ZmPUB76/ZmPUB15*	0.96	1.09	0.88	Segmental	Purify selection	84.18

Note: Ka and Ks indicate the nonsynonymous and synonymous was used to determine the selective pressure after duplication. Ka/Ks = 1 indicates the neutral selection, Ka/Ks > 1 indicates the positive selection, Ka/Ks < 1 indicates the purifying selection. The duplication date (Million years ago, Mya) was calculated by the formula: T = (Ks/2λ) × 10^−6^, where λ = 6.5 × 10^−9^.

## Data Availability

All data analyzed during this study are included in this article and displayed in Appendix A.

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
