# Peer review of "Classification and Expression Profile of the U-Box E3 Ubiquitin Ligase Enzyme Gene Family in Maize (Zea mays L.)"

_plants, 2022, doi:10.3390/plants11192459_

Round 1

Reviewer 1 Report

The research work conducted by Li et al., on “Classification and expression profile of the U-box E3 ubiquitin ligase enzyme gene family in maize (Zea mays L.)” determined the genome wide expression pattern of PUB gene family in maize. Author carried out all the in silico and gene expression analysis in maize. Based on the genome wide they reported 79 PUB genes in maize and categorised to 7 groups. Authors should carry out the following minor revision

Comments to authors:

1. Make the paper to plants journal format. Since it’s in IJMS format.

2. Lots of spell errors are there throughout the MS. Kindly do thorough spell check during revision

3. None of the places the plants, microbial names were italicised

4. Similarly the genes were also not italicised in the MS. Keep italics for genes

5. The legend for figure 6 can be remodified

6. Line no. 494: It should be RNA seq, gene expression profile

7. Line no. 501: should be micromolar

Author Response

Dear Editor,

We would like to resubmit the manuscript titled: “Classification and expression profile of the U-box E3 ubiquitin ligase enzyme gene family in maize (Zea mays L.)" (plants-1890220) to your journal.

Thank you for your careful review of our manuscript. We have revised our manuscript according to your helpful comments and have explained our revisions point by point in this letter (please see below).

We revised the manuscript thoroughly and carefully. All of the changes have been indicated in red color in the revised manuscript.

Best regards,

Guihua Lv

Reviewer reports:

Reviewer 1: The research work conducted by Li et al., on “Classification and expression profile of the U-box E3 ubiquitin ligase enzyme gene family in maize (Zea mays L.)” determined the genome wide expression pattern of PUB gene family in maize. Author carried out all the in silico and gene expression analysis in maize. Based on the genome wide they reported 79 PUB genes in maize and categorised to 7 groups. Authors should carry out the following minor revision.

Comments to authors:

  1. Make the paper to plants journal format. Since it’s in IJMS format.

Re:Thank you for your reminding. We have rearranged the paper according to the plants journal format. Please see the revised manuscript.

  1. Lots of spell errors are there throughout the MS. Kindly do thorough spell check during revision.

Re:Thanks for pointing out our mistake. We have carefully reviewed the manuscript and corrected the spelling errors. Please see the revised manuscript.

  1. None of the places the plants, microbial names were italicised.

Re:Thank you for your reminding. We have checked the format of all plants, microbial names and corrected it to italic. Please see the revised manuscript.  

  1. Similarly the genes were also not italicised in the MS. Keep italics for genes

Re:Thank you for pointing out our mistake. We have checked the format of the genes and changed it to italic. Please see the revised manuscript.

  1. The legend for figure 6 can be remodified

Re: Thank you for your reminder. We have revised it in the manuscript. Please see line 255-256.

The revised contents are as follows:

Line 255-256: Figure 6. A word cloud image of the cis-acting regulatory element of 79 PUB genes in maize. The size and intensity indicate the frequency and occurrence.

  1. Line no. 494: It should be RNA seq, gene expression profile

Re: Thank you for your suggestion. We have revised it in the manuscript. Please see line 611.

The revised contents are as follows:

Line 611: 4.6 RNA-seq and gene expression profile.

  1. Line no. 501: should be micromolar

Re: Thank you for your advice. We have revised it in the manuscript. Please see line 619-620.

The revised contents are as follows:

Line 619-620: Three-leaf stage seedlings were treated with 100 μmol/L ABA, 100 μmol/L IAA, 100 μmol/L GA in 0.1‰ Tween-20, respectively.

Reviewer 2 Report

Dear author(s),

The following points should clearly be corrected and addressed for readers:

Abstract

1.   L15, change plant to plants.

2.   L15, change stress to stresses.

3.   L17, scientific name of plant should be written at the end of sentence as maize (Zea mays L). Please note that scientific name should be italic as you know.

4.   L17-18, change the sentence “In this study, the PUB gene in maize was identified through whole genome screening.” to “In this study, the PUB gene in maize was aimed to identify and classify through whole genome screening”.

5.   L19-20, delete “with MEGA 7, EvolView v2, GSDS 2.0, MEME, MapChart, MCScanX and PlantCARE, respectively.” since readers can see detail in M&M.

6.   L25, change were to was, and plants to plant.

7.   L25, “A close relationship was observed between the monocot plant maize and rice.” Please write what for at the end of sentence. PUB gene??

8.   L25, please a blank before new sentence (after dot).

Keywords

9.   L35, Zea mays could be added.

Introduction

10.           L37 and 44, please use a blank before box brackets or new sentences. Check whole text.

11.           L47, please use full name and then abbreviations. Change “HECT (Homologous to E6-AP COOH-Terminus),” to Homologous to E6-AP COOH-Terminus (HECT),

12.           L48, change RING (Really Interesting New Gene) to Really Interesting New Gene (RING),

13.           L48, change CRLs (Cullin-RING Ligase) to Cullin-RING Ligase (CRLs)

14.           A blank before box bracket??

15.           L48, is PUB plant U-box? If so, please correct.

16.           L50, a blank before box bracket?

17.           L50, please use a reference at the end of sentence who was reported the first U-box gene in yeast.

18.           L53-55, all scientific name should be italicized.

19.           L55, change Truncatula to truncatula.

20.           L60, scientific name should be italicized.

21.           L62, italic??

22.           L65, scientific name of plants should be given in the first mentioned places, and then insist on scientific name or common name. Tabaco???

23.           L66-67, diseases should be written as italic, too. The genus name uppercase and species name with lowercase.

24.           L74, change Rice to In rice,

25.           L78, scientific names should be given in the first mentioned places.

26.           L82, change “we screened….” to we aimed to screen and classify …..”

27.           Aim is enough. Extra explanations should be given in M&M.

Results

28.L93, Zea mays ?? I am not sure italic or not.

29.L115-116, plant names should be italicized.

30.L127-128, plant names should be italicized.

31.L156, plant names should be italicized.

32.L164, write respectively at the end of sentence.

33.L65, change we to were

34.L169, plant names should be italicized.

35.L183-187, plant names should be italicized.

36.L189, plant names should be italicized.

37.L193, plant names should be italicized.

38.L199-200, plant names should be italicized.

39.L216, delete blank before gamma

40.L222, plant names should be italicized.

41.L252, change 8 to Eight

42.L253, change 3 to Three

43.L256, I strongly suggest that passive sentence should be preferred. There are many active sentences.

44.Gene symbols should be italicized.

45.L288, change 24 to Twenty-four

46.L292, change 40 to Forty

47.L296, chsnge 36 to Thirty-six

48.L321, 36?? Sentences should be started with letters

49.L333, P should be italicized.

50. 

Discussion

51.           L343, scientific name of banana?

52.           L349, Kinase or kinase? Cyclophilin or cyclophilin?

53.           L350, plant names should be italicized.

54.           L360, plant names should be italicized.

55.           L365-366, plant names should be italicized.

56.           L432, change Cd to cadmium (Cd) if it is true.

57.           L431, copper copper (Cu)

M&M

58.           L460, plant names should be italicized.

59.           L485, plant names should be italicized.

60.           L499, plant names should be italicized.

61.           L513, change analysis to analyses. It is more than 1

Tables and Figures

62.  I think ZmPUB78 and zmPUB79 were not connected with any chromosomes in Table S1.

63.  Figure 7, too small to view. Could it be given as horizontal.

64.  Figure 8 and 9 too small to view, too.

Author Response

Dear Editor,

We would like to resubmit the manuscript titled: “Classification and expression profile of the U-box E3 ubiquitin ligase enzyme gene family in maize (Zea mays L.)" (plants-1890220) to your journal.

Thank you for your careful review of our manuscript and your encouraging message. We have revised our manuscript according to your helpful comments and have explained our revisions point by point in this letter (please see below).

We revised the manuscript thoroughly and carefully. All of the changes have been indicated in red color in the revised manuscript.

Best regards,

Guihua Lv

Reviewer reports:

Reviewer 2: The following points should clearly be corrected and addressed for readers.

Abstract

  1. L15, change plant to plants.

Re:Thank you for your reminder. We have changed plant to plants. Please see line 16 in the revised manuscript.

  1. L15, change stress to stresses.

Re:Thank you for your advice. We have changed stress to stresses. Please see line 17 in the revised manuscript.

  1. L17, scientific name of plant should be written at the end of sentence as maize (Zea mays L). Please note that scientific name should be italic as you know.

Re:Thank you for your suggestion. We have added the scientific name of maize. Please see line 18 in the revised manuscript.

  1. L17-18, change the sentence “In this study, the PUB gene in maize was identified through whole genome screening.” to “In this study, the PUB gene in maize was aimed to identify and classify through whole genome screening”.

Re:Thank you for your advice. We have changed the sentence. Please see line 19-20 in the revised manuscript.

The revised sentence are as follows:

Line 18-20: In this study, the PUB gene in maize was aimed to identify and classify through whole genome screening.

  1. L19-20, delete “with MEGA 7, EvolView v2, GSDS 2.0, MEME, MapChart, MCScanX and PlantCARE, respectively.” since readers can see detail in M&M.

Re:Thank you for your reminder. We have deleted these words. Please see line 20-21 in the revised manuscript.

The revised contents are as follows:

Line 20-21: Phylogenetic tree, gene structure, conserved motif, chromosome location, gene duplication(GD), synteny and cis-acting regulatory element of PUB member were analyzed.

  1. L25, change were to was, and plants to plant.

Re:Thank you for your reminder. We have revised these words. Please see line 26 in the revised manuscript.

  1. L25, “A close relationship was observed between the monocot plant maize and rice.” Please write what for at the end of sentence. PUB gene??

Re:Thank you for your advice. We have changed the sentence. Please see line 26-27 in the revised manuscript.

The revised contents are as follows:

Line 26-27: A close relationship was observed between the monocot plant maize and rice in PUB gene family.

  1. L25, please a blank before new sentence (after dot).

Re:Thank you for your reminder. We have corrected this mistake. Please see line 26 in the revised manuscript. 

Keywords

  1. L35, Zea mays could be added.

Re:Thank you for your suggestion. We have added the Latin name. Please see line 32 in the revised manuscript.

The revised contents are as follows:

Line 32: Maize (Zea mays L.); U-box E3 (PUB) gene family; Classifification; Expression profile

Introduction

  1. L37 and 44, please use a blank before box brackets or new sentences. Check whole text.

Re:Thank you for your reminder. We have checked whole text and added a blank before box brackets or new sentences. Please see the revised manuscript. 

  1. L47, please use full name and then abbreviations. Change “HECT (Homologous to E6-AP COOH-Terminus),” to Homologous to E6-AP COOH-Terminus (HECT),

Re:Thank you for your advice. We have revised it. Please see line 50-51 in the revised manuscript. 

The revised contents are as follows:

Lin50-51:E3 can be classified into four distinct categories: Homologous to E6-AP COOH-Terminus (HECT), U-box, Really Interesting New Gene (RING), and Cullin-RING Ligase (CRLs).

  1. L48, change RING (Really Interesting New Gene) to Really Interesting New Gene (RING),

Re:Thank you for your advice. We have revised it. Please see line 50-51 in the revised manuscript. 

The revised contents are as follows:

Lin50-51:E3 can be classified into four distinct categories: Homologous to E6-AP COOH-Terminus (HECT), U-box, Really Interesting New Gene (RING), and Cullin-RING Ligase (CRLs).

  1. L48, change CRLs (Cullin-RING Ligase) to Cullin-RING Ligase (CRLs)

Re:Thank you for your advice. We have revised it. Please see line 50-51 in the revised manuscript. 

The revised contents are as follows:

Lin50-51:E3 can be classified into four distinct categories: Homologous to E6-AP COOH-Terminus (HECT), U-box, Really Interesting New Gene (RING), and Cullin-RING Ligase (CRLs).

  1. A blank before box bracket??

Re:Thank you for your reminder. We have added a blank before box bracket. Please see the revised manuscript. 

  1. L48, is PUB plant U-box? If so, please correct.

Re:Thank you for your suggestion. In this paper, PUB is as an abbreviation of the U-box E3. We added the description in the abbreviations part.

  1. L50, a blank before box bracket?

Re:Thank you for your reminder. We have added a blank before box bracket. Please see line 54 in the revised manuscript. 

  1. L50, please use a reference at the end of sentence who was reported the first U-box gene in yeast.

Re:Thank you for your advice. We have added the reference in the manuscript. Please see line 55, 724,725 in the revised manuscript. 

The revised contents are as follows:

Lin724-725: Koegl, M.; Hoppe, T.; Schlenker, s.; Ulrich, h.d.; Mayer, t.u.; jentsch, s. A novel ubiquitination factor, E4, is involved in multiubiquitin chain assembly. Cell 1999, 96, 635-644. https://doi.org/10.1016/s0092-8674(00)80574-7.

  1. L53-55, all scientific name should be italicized.

Re:Thank you for your reminder. We have corrected the scientific name in  italics. Please see line 58-62 in the revised manuscript.

The revised contents are as follows:

Line 58-62:Till date, the PUB gene family was detected in numerous species, namely, Arabidopsis (Arabidopsis thaliana, 64), rice (Oryza sativa, 77),  chinese cabbage (Brassica rapa ssp. Pekinesis, 101), soybean (Glycine max, 125), barley (Hordeum vulgare L., 67), tomato (Lycopersicon esculentum, 62), citrus (Citrus clementina, 56), grape (Vitis vinifera, 56), cotton (Gossypium raimondii, 93), and Medicago (Medicago truncatula, 41).

  1. L55, change Truncatula to truncatula.

Re:Thank you for your reminder. We have revised it. Please see line 62 in the revised manuscript.

  1. L60, scientific name should be italicized.

Re:Thank you for your reminder. We have corrected the scientific name in  italics. Please see line 69 in the revised manuscript.

  1. L62, italic??

Re:Thank you for your reminder. We have corrected the scientific name in italics. Please see line 71 in the revised manuscript.

  1. L65, scientific name of plants should be given in the first mentioned places, and then insist on scientific name or common name. Tabaco???

Re:Thank you for your suggestion. We have added the scientific name of tabacco . Please see line 74-75 in the revised manuscript.

The revised contents are as follows:

Line 74-75: tobacco (Nicotiana tabacum) and tomato

  1. L66-67, diseases should be written as italic, too. The genus name uppercase and species name with lowercase.

Re:Thank you for your reminder. We have revised it. Please see line 77-78 in the revised manuscript.

  1. L74, change Rice to In rice,

Re:Thank you for your reminder. We have revised it. Please see line 88 in the revised manuscript.

  1. L78, scientific names should be given in the first mentioned places.

Re:Thank you for your suggestion. We have added the scientific name. Please see line 92 in the revised manuscript.

The revised contents are as follows:

Line 92: potato (Solanum tuberosum) and Arabidopsis

  1. L82, change “we screened….” to we aimed to screen and classify …..”

Re:Thank you for your suggestion. We have changed the sentence. Please see line 98-100 in the revised manuscript.

The revised contents are as follows:

Line 98-100: Herein, we aimed to screen and classify the PUB gene family in maize and performed systematic and comprehensive analyses.

  1. Aim is enough. Extra explanations should be given in M&M.

Re:Thank you for your advice. We have deleted some sentences. Please see line 98-101 in the revised manuscript.

Results

  1. L93, Zea mays ?? I am not sure italic or not.

Re:Thank you for your reminder. We used the common name in this sentence. Please see line 105 in the revised manuscript.

The revised contents are as follows:

Line105:79 putative PUB genes were screened with HMMER, using default parameters and significant e-3 value against the maize genome

  1. L115-116, plant names should be italicized.

Re:Thank you for your reminder. We corrected these words. Please see line 135 in the revised manuscript.

The revised contents are as follows:

Line135:There were similar proportions surveyed in Arabidopsis, rice, barley and citrus.

  1. L127-128, plant names should be italicized.

Re:Thank you for your reminder. We corrected these words. Please see line 146 in the revised manuscript.

The revised contents are as follows:

Line146:U-box domain sequences of rice, Arabidopsis and maize were performed using MEGA 7

  1. L156, plant names should be italicized.

Re:Thank you for your reminder. We used the common name in this sentence. Please see line 174 in the revised manuscript.

  1. L164, write respectively at the end of sentence.

Re:Thank you for your advice. We added “respectively” in the end of sentence. Please see line 186 in the revised manuscript.

  1. L65, change we to were

Re:Thank you for your advice. We corrected it. Please see line 186 in the revised manuscript.

  1. L169, plant names should be italicized.

Re:Thank you for your reminder. We used the common name in this sentence. Please see line 191 in the revised manuscript.

  1. L183-187, plant names should be italicized.

Re:Thank you for your reminder. We revised the plant names. Please see line 210-214 in the revised manuscript.

  1. L189, plant names should be italicized.

Re:Thank you for your advice. We revised the plant names. Please see line 216 in the revised manuscript.

  1. L193, plant names should be italicized.

Re:Thank you for your advice. We revised the plant names. Please see line 220 in the revised manuscript.

  1. L199-200, plant names should be italicized.

Re:Thank you for your advice. We revised the plant names. Please see line 227-228 in the revised manuscript.

  1. L216, delete blank before gamma

Re:Thank you for your advice. We delete blank before gamma. Please see line 224 in the revised manuscript.

  1. L222, plant names should be italicized.

Re:Thank you for your advice. We revised the plant names. Please see line 255 in the revised manuscript.

  1. L252, change 8 to Eight

Re:Thank you for your advice. We have changed 8 to Eight. Please see line 295 in the revised manuscript.

  1. L253, change 3 to Three

Re:Thank you for your advice. We have changed 3 to Three. Please see line 296 in the revised manuscript.

  1. L256, I strongly suggest that passive sentence should be preferred. There are many active sentences.

Re:Thank you for your suggestion. We have changed this sentence. Please see line 300-301 in the revised manuscript.

The revised contents are as follows:

Line 300-301: The promoter regions of the highly expressed genes were further analyzed.

  1. Gene symbols should be italicized.

Re:Thank you for your reminder. We have modified the gene symbols with italics. Please see the revised manuscript.

  1. L288, change 24 to Twenty-four

Re:Thank you for your advice. We have changed 24 to twenty-four. Please see line 341 in the revised manuscript.

  1. L292, change 40 to Forty

Re:Thank you for your advice. We have changed 40 to Forty. Please see line 348 in the revised manuscript

  1. L296, chsnge 26 to Twenty-six

Re:Thank you for your advice. We have changed 26 to twenty-six. Please see line 349 in the revised manuscript

  1. L321, 32?? Sentences should be started with letters

Re:Thank you for your advice. We have changed 32 to thirty-two. Please see line 353 in the revised manuscript

  1. L333, P should be italicized.

Re:Thank you for your advice. We have revised it. Please see line 401 in the revised manuscript

Discussion

  1. L343, scientific name of banana?

Re:Thank you for your reminder. We have added the scientific name of banana. Please see line 413-414 in the revised manuscript

The revised contents are as follows:

Line 413-414: in rice, soybean, barley, citrus and banana (Musa paradisiaca)

  1. L349, Kinase or kinase? Cyclophilin or cyclophilin?

Re:Thank you for your reminder. We have revised it. Please see line 421-422 in the revised manuscript

The revised contents are as follows:

Line 421-422:both the U-box domain and several other domains, such as ARM repeat (27), kinase (14), WD40 repeat (2), MIF4G (2), cyclophilin (1), UFD2 (1) and TPR domains (1)

  1. L350, plant names should be italicized.

Re:Thank you for your reminder. We have motified the plant name in italics . Please see line 423 in the revised manuscript

  1. L360, plant names should be italicized.

Re:Thank you for your reminder. We have motified the plant name in italics. Please see line 436 in the revised manuscript

  1. L365-366, plant names should be italicized.

Re:Thank you for your reminder. We have motified the plant name in italics. Please see line 443-444 in the revised manuscript

  1. L432, change Cd to cadmium (Cd) if it is true.

Re:Thank you for your reminder. We have changed Cd to cadmium (Cd). Please see line 533 in the revised manuscript

  1. L431, copper copper (Cu)

Re:Thank you for your reminder. We have changed copper to copper (Cu). Please see line 532 in the revised manuscript

M&M

  1. L460, plant names should be italicized.

Re:Thank you for your reminder. We have revised it. Please see line 571 in the revised manuscript

The revised contents are as follows:

Line 571:The maize genome sequences

  1. L485, plant names should be italicized.

Re:Thank you for your reminder. We have revised it. Please see line 600 in the revised manuscript

The revised contents are as follows:

Line 600:Arabidopsis, maize and rice

  1. L499, plant names should be italicized.

Re:Thank you for your reminder. We have revised it. Please see line 617 in the revised manuscript

The revised contents are as follows:

Line617:Zea mays cv. B73

  1. L513, change analysis to analyses. It is more than 1

 Re:Thank you for your reminder. We have revised it. Please see line 635 in the revised manuscript

The revised contents are as follows:

Line635: Statistical analyses

Tables and Figures

  1. I think ZmPUB78 and zmPUB79 were not connected with any chromosomes in Table S1.

Re:Thank you for your reminder. In total, 77 ZmPUB genes (97.4%) were mapped to 10 chromosomes, and 2 ZmPUB genes were located in contig 24 and 182. We have revised the Table S1. Please see Table S1.

  1. Figure 7, too small to view. Could it be given as horizontal.

Re:Thank you for your suggestion. We redrew the Figure 7. Please see line 275 in the revised manuscript.

  1. Figure 8 and 9 too small to view, too.

Re:Thank you for your suggestion. We redrew the Figure 8 and 9. Please see line 308 and 332 in the revised manuscript.